# Change in Descriptive Kinematic Parameters of Patients with Patellofemoral Instability When Compared to Individuals with Healthy Knees—A 3D MRI In Vivo Analysis

**DOI:** 10.3390/jcm12051917

**Published:** 2023-02-28

**Authors:** Markus Siegel, Philipp Maier, Elham Taghizadeh, Andreas Fuchs, Tayfun Yilmaz, Hans Meine, Hagen Schmal, Thomas Lange, Kaywan Izadpanah

**Affiliations:** 1Department of Orthopedic Surgery and Traumatology, Freiburg University Hospital, Albert Ludwigs University Freiburg, Hugstetter Straße 55, 79106 Freiburg, Germany; 2Fraunhofer Institute for Digital Medicine MEVIS, Max-von-Laue-Str. 2, 28359 Bremen, Germany; 3Department of Orthopedic Surgery, University Hospital Odense, Sdr. Boulevard 29, 5000 Odense C, Denmark; 4Division of Medical Physics, Department of Diagnostic and Interventional Radiology, Medical Center—University of Freiburg, Faculty of Medicine, University of Freiburg, Killianstrasse 5a, 79106 Freiburg, Germany

**Keywords:** patellofemoral, PFI, patellofemoral instability, kinematic, dynamic MRI

## Abstract

Background: Patellofemoral instability (PFI) leads to chronic knee pain, reduced performance and chondromalacia patellae with consecutive osteoarthritis. Therefore, determining the exact patellofemoral contact mechanism, as well as the factors leading to PFI, is of great importance. The present study compares in vivo patellofemoral kinematic parameters and the contact mechanism of volunteers with healthy knees and patients with low flexion patellofemoral instability (PFI). The study was performed with a high-resolution dynamic MRI. Material/Methods: In a prospective cohort study, the patellar shift, patella rotation and the patellofemoral cartilage contact areas (CCA) of 17 patients with low flexion PFI were analyzed and compared with 17 healthy volunteers, matched via the TEA distance and sex, in unloaded and loaded conditions. MRI scans were carried out for 0°, 15° and 30° knee flexion in a custom-designed knee loading device. To suppress motion artifacts, motion correction was performed using a moiré phase tracking system with a tracking marker attached to the patella. The patellofemoral kinematic parameters and the CCA was calculated on the basis of semi-automated cartilage and bone segmentation and registrations. Results: Patients with low flexion PFI showed a significant reduction in patellofemoral CCA for 0° (unloaded: *p* = 0.002, loaded: *p* = 0.004), 15° (unloaded: *p* = 0.014, loaded: *p* = 0.001) and 30° (unloaded: *p* = 0.008; loaded: *p* = 0.001) flexion compared to healthy subjects. Additionally, patients with PFI revealed a significantly increased patellar shift when compared to volunteers with healthy knees at 0° (unloaded: *p* = 0.033; loaded: *p* = 0.031), 15° (unloaded: *p* = 0.025; loaded: *p* = 0.014) and 30° flexion (unloaded: *p* = 0.030; loaded: *p* = 0.034) There were no significant differences for patella rotation between patients with PFI and the volunteers, except when, under load at 0° flexion, PFI patients showed increased patellar rotation (*p* = 0.005. The influence of quadriceps activation on the patellofemoral CCA is reduced in patients with low flexion PFI. Conclusion: Patients with PFI showed different patellofemoral kinematics at low flexion angles in both unloaded and loaded conditions compared to volunteers with healthy knees. Increased patellar shifts and decreased patellofemoral CCAs were observed in low flexion angles. The influence of the quadriceps muscle is diminished in patients with low flexion PFI. Therefore, the goal of patellofemoral stabilizing therapy should be to restore a physiologic contact mechanism and improve patellofemoral congruity for low flexion angles.

## 1. Introduction

Patellofemoral instability (PFI) has received a great deal of attention lately due to the assumption that it is a cause of anterior knee pain [1]. According to the current state of knowledge, PFI can be considered a significant factor in knee pain, with an overall prevalence of 6 per 100,000 [2,3]. The highest prevalence is among young, athletic men during physical activity, with a prevalence of 30 per 100,000 [2,4]. PFI leads to chronic knee pain, reduced performance and chondromalacia patellae with consecutive osteoarthritis [5]. Due to the high prevalence and the direct association of PFI with patellofemoral pain syndrome (PFPS) and in the case of recurrent dislocation osteoarthritis, it is necessary to identify the exact pathomechanism before drawing any clinical conclusions [6]. Several factors (such as reduced strength of the quadriceps muscle, an altered Q-angle, patellofemoral malalignment, dysplasia of the femoral trochlea and the medial patellofemoral ligament) are considered important factors in the pathogenesis of PFI [7,8].

Especially in low knee flexion angles (0–30°), soft tissue structures, such as the medial patellofemoral ligament (MPFL) and the quadriceps muscle, play an eminent role in patellar tracking [1,9]. PFI is closely connected to altered patellofemoral kinematics and altered cartilage contact mechanisms [10,11]. In particular, little is known about the influence of the quadriceps on patellofemoral kinematics and patellofemoral contact. To analyze the patellofemoral interaction, high-quality in vivo dynamic MRI examinations are becoming increasingly important [12,13,14].

Nevertheless, the exact pathomechanism in vivo has not yet been fully delineated, nor has the role of quadriceps muscles in patients with PFI been fully elucidated.

This study aims to describe the patellofemoral kinematic and contact mechanism of patients with low flexion PFI, then compare them to a control group with healthy knees in unloaded and loaded situations via high-resolution dynamic in vivo MR imaging, using semi-automated 3D meshing techniques.

We hypothesize that patients with low flexion PFI have different patellofemoral kinematics compared to individuals with healthy knees. In addition, we hypothesize that quadriceps activation alone might not significantly influence these parameters.

## 2. Material/Methods

Study population and diagnostic criteria:

In this prospective matched pair cohort study, 17 healthy volunteers and 17 patients with patellar instability at low flexion angle before surgical treatment were investigated. The inclusion criteria for patients were a clinically apparent low flexion patellofemoral instability from 0° to 30° knee flexion, with indications of stabilizing surgery, and no history of knee joint surgery. Patients and volunteers who had knee surgery, implanted metallic material in the region of interest, retro-patellar osteoarthritis, chondromalacia patellae or were pregnant were excluded a priori. Only volunteers without a history of knee pain were included.

The diagnosis of low flexion PFI was based on clinical symptoms of patellofemoral pain, subjective clinical instability in low flexion angles (0–30°) and a history of recurrent patellofemoral luxation. Each patient was diagnosed by a clinical examination, X-ray and MRI of the knee joint. The patients were recruited from a waiting list for stabilizing surgery via isolated MPFL reconstruction. Potential probands were contacted by phone or asked for consent during ambulant consultation. Volunteers were enrolled using advertising or personal contacts.

A total of 20 patients with low flexion PFI were considered potential probands. These could be recruited for this study and were examined by the MRI protocol. Three of these patients could not be included, two because of technical issues and one because of undiagnosed claustrophobia. From this cohort, 17 complete MRI datasets could be obtained for further processing. Twenty-one healthy volunteers were recruited for the control group and were also examined for study purposes according to the MRI protocol. Two datasets had to be excluded because of technical problems. Finally, 17 complete datasets of this cohort could be generated for further analysis. The sample size of 17 was ultimately based on the complete records of the patellofemoral unstable patient cohort (after dropouts). In order to generate comparable groups, matching was performed using both gender and transepicondylar distance (TEA distance) as a parameter of knee size, resulting in a matched-pair cohort of 17 patients with PFI and 17 volunteers (see Figure 1).

### 2.1. Participants

Patients with PFI: Mean patient age was 26.47 ± 7.67 years (range, 18–49). Their mean height was 174.59 ± 8.73 cm (range 160–189), and their mean weight was 71.35 ± 10.88 kg (range 55–100). Ten participants (58.8%) were female, and 7 were male (41.2%). The mean TEA distance of Patients with PFI was 77.13 ± 6.05 cm.

Volunteers with healthy knees: Average age of 30.00 ± 5.81 years (range, 22–41). Their mean height was 176.24 ± 9.50 cm (range, 168–195), and their mean weight was 75.47 ± 18.27 kg (range, 59–137). Ten women (58.8%) and 7 men (41.2%) participated. The mean TEA distance of healthy volunteers was 79.63 ± 7.58 cm; compared to patients with PFI, there is a *p* = 0.245. Patients’ demographics are shown in Table 1.

The study was approved by the Institutional Review Board (the Freiburg University ethics committee, ID 443/16), and all subjects gave written informed consent before participation. The study was registered in the German Trials Register (DRKS 00029213)

### 2.2. MRI Setup and Protocol

The study was performed on a Magneton Trio 3 T system (Siemens Healthineers, Erlangen, Germany), combined with an 8-channel multipurpose coil (NORAS MRI products, Germany), which was attached to the thigh via a hook-and-loop fastener. A special MR-compatible pneumatic loading apparatus (Figure 2) designed for the lower limb was used to apply axial in situ knee loading in the range of 0–500 N from the console room leading to a consecutive muscular activation of the quadriceps muscles.

As a safety precaution, the subjects were given an emergency stop button, which enabled them to immediately release the load. The subjects were fastened on the scanner bed 3 by a weight-lifting belt, with the investigated leg placed on the sliding carriage of the loading device. Measurements were performed with the knee angled at 0°, 15° and 30°, first without and then with a knee load of 50 N. We chose a load of 50 N in order to activate the muscles without generating enormous motion artifacts in order to gain a robust and reproducible method.

A 3D turbo-spin echo (TSE) protocol with GRAPPA parallel imaging acceleration by a factor of 2 and an isotropic resolution of 0.5 mm was applied for the MRI scans. Further scan parameters were TR = 1.8 s, TE = 59 ms, receiver bandwidth = 504 Hz/Px, and scan duration = 6:20 min. The 3D measurement volume was positioned so that it covered the whole patellofemoral joint. For prospective motion correction, we used a moiré phase tracking (MPT) system (Metria Innovation, Milwaukee, WI, USA). The system consisted of a single in-bore camera and a single tracking marker, which was attached to the subject’s kneecap [15]. The tracking system detected translational and rotational movement of the marker at a rate of 80 frames per second. With the optically tracked motion data, a real-time update of the MRI measurement volume was performed before every excitation pulse. Furthermore, inter-scan motion correction (position locking) ensured that all MRI scans were performed with the initially planned field of view (FOV).

Data processing and quantification were performed with the image processing platform SATORI, developed by Fraunhofer MEVIS on the basis of MeVisLab. The bones and cartilages of the patellofemoral joint were manually segmented by experts in the images acquired in knee extension without loading. Three-dimensional meshes of each structure were generated from the segmented images (Figure 3) and were transferred via image registration to all other datasets acquired at different flexion and loading situations.

The image registration was performed for each bone individually, optimizing a normalized gradient fields (NGF) distance measure on the MRI data, restricted to a region around the respective bone. Based on the assumption that the bones do not deform significantly, the result of this bone alignment is individual rigid transformation matrices for all bones and thus articulated mesh models that were then used to analyze the CCA and the patellofemoral kinematics.

### 2.3. Parameter

CCA: The determination of the CCA is based on the 3D surface mesh generated and defined as 2 opposing cartilage surfaces with an inter-cartilage distance below 1 mm. This definition proved to yield contiguous cartilage surfaces without holes across the whole dataset [16].

Patellar shift: In literature, the patellar shift is defined as a transverse movement of the patellar center point along a mediolateral axis of the femur or the patella itself [10,17]. We have defined the patellar shift as the patella movement along the transcondylar axis (TCA, red line in Figure 3). Medial shift is categorized as negative, whereas lateral shift is categorized as positive displacement. We determined the TCA, based on the 3D meshes of bone and cartilage, for each study participant.

Patellar rotation: The patellar rotation is a movement of the patella around an anterior to posterior axis of the femur or patella [18]. We defined the Patellar rotation as the angle between the medial-lateral axis of the patella and the femoral transepicondylar axis (TEA) projected onto the coronal plane of the femur. This movement can be subdivided into positive and negative rotations. Lateral or positive rotation is defined as an outwards rotation of the distal patella pole. At internal rotation or negative rotation, the distal pole of the patella performs inward rotation [10,18,19]. To provide reliable comparability, we defined the patella position at unloaded 0° knee flexion as a starting position (0° rotation) in both PFI patients and healthy volunteers.

### 2.4. Statistics

We determined mean values and standard deviations (SD) for both cohorts. For statistical evaluation, Wilcoxon rank-sum tests were used. Differences with *p* < 0.05 were considered significant. Statistical analyses were conducted using the Python Scipy library (3.6.5). The descriptive statistics are presented in mean values and SD. All analyses were exploratory in nature. As a result, *p*-values and 95% confidence intervals were not corrected for multiple comparisons and inferences drawn from them may not be reproducible.

## 3. Results

In total, the complete data sets of 17 patients with low flexion PFI and 17 datasets of volunteers with healthy knees could be included.

### 3.1. CCA

In both cohorts, the CCA increased in the early range of flexion. For patients with low flexion PFI, we observed a mean increase of 48.12 mm^2^ from extension to 15° flexion (*p* = 0.001) and a mean increase of 172.34 mm^2^ from 15° to 30° flexion (*p* ≤ 0.001). Absolute values can be found in Table 2.

Volunteers with healthy knees showed a mean increase of 60.45 mm^2^ from extension to 15° flexion (*p* = 0.004) and a mean increase of 191.97 mm^2^ from 15° to 30° flexion (*p* ≤ 0.001).

The mean CCA difference for 0° flexion was 60.87 mm^2^ (*p* = 0.002) when PFI patients were compared with healthy volunteers. For 15° knee flexion, the difference was 73.20 mm^2^ (*p* = 0.012), and the mean difference was 92.83 mm^2^ (*p* = 0.007) for 30° knee flexion. Consequently, there is a statistically significant CCA reduction in unloaded conditions when patients with PFI were compared to volunteers with healthy knees (see Figure 4).

In loaded conditions (50 N), there was a mean CCA increase of 36.36 mm^2^ from extension to 15° flexion (*p* = 0.001) and a mean increase of 192.46 mm^2^ from 15° to 30° flexion (*p* < 0.001) for patients with PFI. Volunteers with healthy knees showed a mean CCA increase of 65.34 mm^2^ from extension to 15° flexion (*p* = 0.001) and a mean increase of 237.86 mm^2^ from 15° to 30° flexion (*p* ≤ 0.001). The mean CCA difference in loaded situations for 0° flexion was 60.06 mm^2^ (*p* = 0.005) when PFI patients were compared with healthy volunteers. For 15° knee flexion, the difference was 89.04 mm^2^ (*p* = 0.002), and the mean difference was 134.44 mm^2^ (*p* = 0.001) for 30° knee flexion. There was also a statistically significant CCA reduction in loaded conditions when patients with PFI were compared to volunteers with healthy knees (Figure 4). An overview of the parameters assessed is presented in Table 2. Table 3 illustrates the influence of quadriceps activation within the two cohorts.

### 3.2. Patella Shift

For Patients with low flexion PFI, in unloaded situations, there was a mean increase of the patella shift of 0.21 mm from extension to 15° flexion (*p* = 0.026) and a mean increase of 0.18 mm from 15° to 30° flexion (*p* = 0.005). Volunteers with healthy knees showed a mean increase of 0.39 mm from extension to 15° flexion (*p* = 0.020) and 0.49 mm from 15° to 30° flexion (*p* = 0.002). The comparison of patients with PFI and volunteers is illustrated in Figure 5A. Consequently, there was a statistically significant increase of the patella shift in unloaded conditions when patients with PFI were compared to volunteers with healthy knees from 0° to 15° knee flexion and a statistical trend in 30° flexion.

In loaded conditions (50 N), patients with PFI revealed a mean patella shift increase of 0.2 mm from extension to 15° flexion (*p* = 0.002) and a mean increase of 0.35 mm from 15° to 30° flexion (*p* = 0.019). Volunteers with healthy knees showed a mean change of the patellar shift of 0.52 mm from extension to 15° flexion (*p* = 0.006) and a mean increase of 0.62 mm from 15° to 30° flexion (*p* ≤ 0.002). The comparison between both cohorts is shown in Figure 5B. Consequently, the increase of lateralization in loaded conditions when patients with PFI were compared to volunteers with healthy knees is statistically significant in all knee positions (extension, 15° flexion and 30° flexion).

### 3.3. Patella Rotation Angle

Patients with PFI showed a mean patella rotation angle increase of 3.80° from extension to 15° flexion (*p* ≤ 0.001) and a mean increase of 2.65° from 15° to 30° flexion (*p* = 0.121) in unloaded conditions. For volunteers with healthy knees, we observed a mean increase in Patella rotation angle of 1.39° from extension to 15° flexion (*p* ≤ 0.001) and of 0.59° from 15° to 30° flexion (*p* = 0.149). The mean patella rotation angle difference for 0° flexion was set to 0°, as we used this patellar position as a starting value for further measurements and to improve the comparability of the individual values. There was no statistically significant increase in the patella rotation angle in unloaded conditions when patients with PFI were compared to volunteers with healthy knees, with only a statistical trend for 30° flexion (see Figure 6A).

In loaded conditions (50 N), a mean increase of patella rotation of 1.92° from 0° flexion (*p* = 0.062) to 15° flexion, and a mean increase of 2.62° from 15° to 30° flexion (*p* = 0.234) was observed in the PFI group. Volunteers with healthy knees showed a mean patella rotation angle increase of 1.54° from 0° (*p* = 0.011) to 15° flexion and 1.49° from 15° to 30° flexion (*p* = 0.163) under load. Comparing the load-induced patella rotation angles at 0° flexion, a significantly larger patella rotation was observed for patients with PFI compared to volunteers with healthy knees. Table 3 illustrates the influence of quadriceps activation within the two cohorts on the patella rotation angle.

## 4. Discussion

The main findings of this study confirm that patients with low flexion PFI show different patellofemoral kinematics in low flexion angles (0–30°) compared to volunteers with healthy knees. There is a reduced CCA in all flexion positions in patients with PFI, both with and without quadriceps activation; there is a larger patellar shift (in all flexion angles) and a greater patella rotation (in 0°, loaded) in patients with PFI. The influence of the quadriceps muscle is diminished, especially when considering the muscular influence on patellofemoral congruency at 30° flexion.

In agreement with previous in vitro cadaver studies and a few MRI studies, we can conclude that the patellofemoral contact area of patients with low flexion PFI is significantly reduced when compared with the knees of healthy volunteers in early knee-flexion (0–30°) in loaded and unloaded situations [11,20,21,22]. There is also congruent data that reveals that the patellofemoral CCA increases during early flexion in loaded and unloaded situations in healthy and patellofemoral unstable knees [11,16]. When comparing the CCA of patients with PFI, with and without quadriceps activation (load), there is no significant difference in any of the three flexion positions. In volunteers with healthy knee joints, there is also no significant change in CCA in 0° or 15° flexion. At 30°, however, a significant increase is seen under quadriceps activation with 50 N.

As many authors have already mentioned, the role of active and passive soft tissue stabilizers (such as the quadriceps and the MPFL) is to act as a unit, guiding the patella at low flexion angles (0–30°), with an increased influence from the quadriceps muscle in the course of early flexion [10,23]. Quadriceps activation is the main cause of the contact pressure between the patella and trochlea, giving rise to sufficient CCA as the patella engages into the trochlear groove at about 30° knee flexion [24]. There is, therefore, a difference in the CCA seen in the comparison of loaded and unloaded conditions at 30° knee flexion of 2.50% in patients with PFI, whereas the CCA increase in healthy volunteers in the aforementioned setting is 11.21%. For extension and 15° flexion, the load-induced CCA change in both groups is almost the same. We attribute this observation to the fact that, in extended knees, the influence of soft tissue tension is the lowest, and the influence of the quadriceps muscle increases over the first 30° of knee flexion [8]. With increasing flexion, the difference in the CCA (between loaded and unloaded situations) in both healthy and PFI patients increases, as mentioned before. This might be attributed to the increasing influence of the soft tissue tension, which becomes apparent as the knee begins flexion.

Concerning the transverse patellar displacement (patellar shift) with regard to the transcondylar axis (TCA), this study gives indications that the patella of patients with PFI is significantly lateralized compared to the knees of healthy volunteers. For both patients with PFI and healthy volunteers, the patella shifts medially in the course of early flexion angles (0–30°), confirming previously described observations [17,19,25]. However, several studies have shown that with increasing knee flexion, the patella first shifts laterally, followed by a temporary medial shift [15,26].

From our data, it might be deduced that patients with PFI have a significantly increased patellar lateralization in loaded and unloaded conditions when compared to volunteers with healthy knees in the range from 0° to 30° knee flexion when compared to healthy volunteers, which in turn might contribute to the aforementioned reduction in patellofemoral CCA and negatively affects patellofemoral congruity in early degrees of knee flexion.

The third parameter investigated in this study was the patellar rotation angle. We established our patella zero point at 0° knee flexion in unloaded conditions for patients and volunteers. We, therefore, do not have comparative values between healthy volunteers and patients with PFI for the unloaded extension position. However, especially at 0° flexion (extension), the differences concerning patellar rotation between volunteers and PFI patients with quadriceps activation differ significantly. In agreement with other studies, an increased lateral rotation of the patella in both cohorts in loaded and unloaded conditions can be observed in this study in the early range of flexion (0–15°, with and without load) [27]. Some authors have reached the conclusion that the patella rotates only in the range from −1° to 2° in a range from 0° to 30° knee flexion [15,17]. Our data emphasize that knee flexion significantly influences patella rotation, at least in the range of 0–15° flexion, in a mean range of about 6.5° in both cohorts. When looking at the differences in patellar rotation between volunteers with healthy knee joints and patients with PFI, there is an increased external rotation of the patella in patellofemorally unstable patients only in loaded extension.

PFI patients show different patellar kinematics in loaded and unloaded conditions compared to healthy individuals. Patients with PFI show abnormal patella lateralization (patella shift) in all flexion angles and a consecutively decreased CCA in early flexion (0–30°). Furthermore, the influence of the quadriceps muscle is diminished in patients with PFI.

From these observations, we can conclude that the soft tissue restraints and the muscular restraining abilities in the patellofemoral contact mechanisms in patients with low flexion PFI are altered, leading to increased lateralization and disproportionally reduced CCA. One potential cause for the reduced influence of quadriceps activation on the patellofemoral contact mechanism in patients with low flexion PFI is a differently orientated compressing force vector of the quadriceps in patients with PFI to healthy volunteers, which usually improves congruity [24]. Due to the assumption that an altered cartilage contact mechanism, the reduced patella femoral congruity and the consecutively reduced CCA cause premature osteoarthritis of the patellofemoral joint due to joint misload, we consider the restoration of patellofemoral congruency and therefore a physiological CCA in early flexion to be the core of patellofemoral stabilizing therapies for patients with low flexion PFI [28,29].

Nevertheless, the isolated role of the CCA has not been conclusively answered. The efficacy of conservative therapy, which aims at increasing the CCA and thus improving congruency, must be questioned based on the results we observed on average with a missing effect of quadriceps activation on the CCA. Nevertheless, some patients showed an increase in CCA comparable to healthy volunteers under quadriceps activation.

Eventually, those patients without relevant quadriceps influence on the patellofemoral CCA are more suitable for a primary surgical patellofemoral stabilizing approach (e.g., MPFL reconstruction with or without concomitant procedures), while patients with an effect of quadriceps activation on the CCA might profit from conservative approaches.

Consequently, the influence and measurability of muscular patellofemoral stabilization could be suitable as a predictive factor to determine the best patient-individualized therapeutic approach. Further prospective in vivo studies are needed to validate this hypothesis and to identify additional factors that influence the therapeutic outcome of patients with low flexion PFI.

In this study, we used a robust method that enabled us to assess and compare the kinematic parameter of patients with PFI and volunteers with no history of knee pain or instability in vivo. We consider the individual in vivo assessment of the patellofemoral contact mechanism essential for planning the best therapeutic approach in patients with patellofemoral pathologies. The results obtained in this study will contribute to the further understanding of the pathomechanism of patellofemoral instability.

### Limitations

One of the limitations of this study is the study population. The results should be considered exploratory and may differ for larger cohorts. To ensure relative homogenization of the trial cohort in the heterogeneous condition of PFI, only patients with low flexion PFI were included, and all patients were recruited from the waiting list for patellofemoral stabilizing surgery. It is possible that results may vary in patients who were not preselected in this regard. Due to the small number of study participants, we did not perform subgroup analyses (e.g., based on trochlear shapes, TTTG/TTPCL, valgus leg axis, or patella height). This is of great interest to draw further clinical conclusions and is the subject of current research projects/follow-up projects.

As the study used a closed-bore MRI scanner unit with a diameter of 24 inches, the angles of knee flexion which could be investigated were limited. However, the range of 0 to 30° flexion did not lead to any limitations in our assessment. Knee MRI with loading under flexion is very prone to give rise to motion artifacts. Other authors have also faced the problem of motion artifacts, especially during weight-bearing or consecutive quadriceps loading, leading to measurement inaccuracies [22]. We addressed this problem by using prospective motion correction via an in-bore camera, which suppressed the artifacts and led to accurate measurements [30].

MR imaging is based on static scanning, which can only acquire a limited number of images at certain knee flexion angles. The full understanding of patellofemoral kinematics between this limited dataset could only be estimated with the help of interpolation and thus might have led to inaccuracies [31].

## 5. Conclusions

Patients with PFI have significantly different patellofemoral kinematics and cartilage contact mechanisms, both in unloaded and loaded conditions, to healthy volunteers. PFI patients show increased patellar lateralization in all flexion positions, with and without loading. The CCA is significantly lower when compared to healthy participants. The influence of the quadriceps muscle is diminished in patients with PFI.

## Figures and Tables

**Figure 1 jcm-12-01917-f001:**
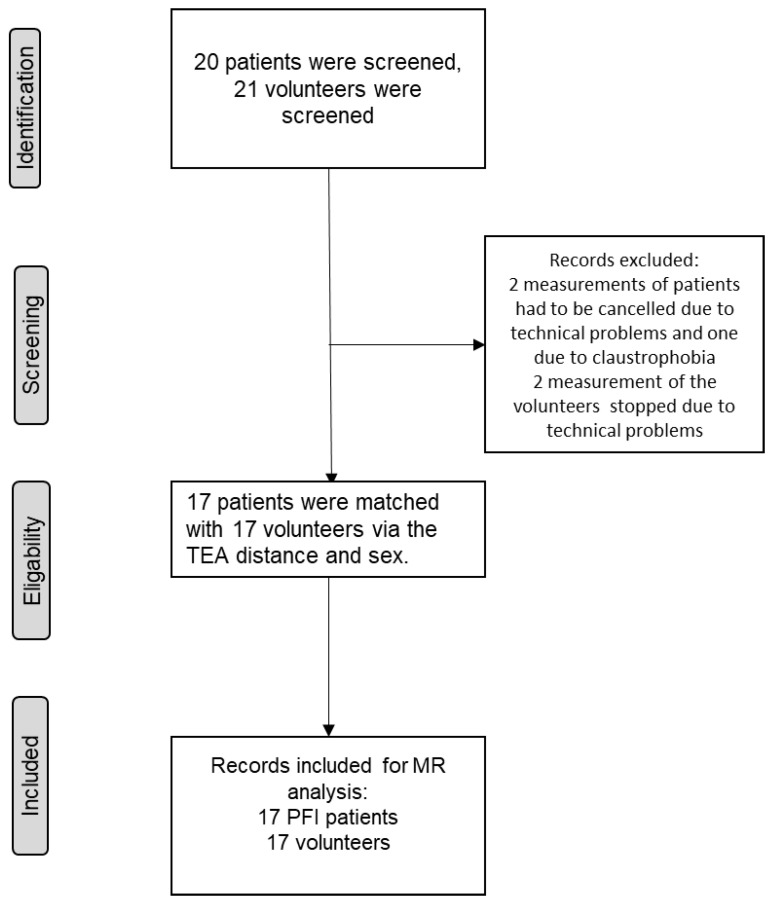
Flow diagram of the selection process according to the STROBE guidelines.

**Figure 2 jcm-12-01917-f002:**
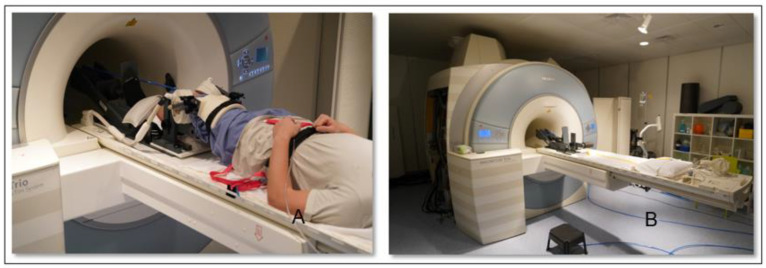
The pneumatic loading device (**A**) and the complete MRI Setup (**B**).

**Figure 3 jcm-12-01917-f003:**
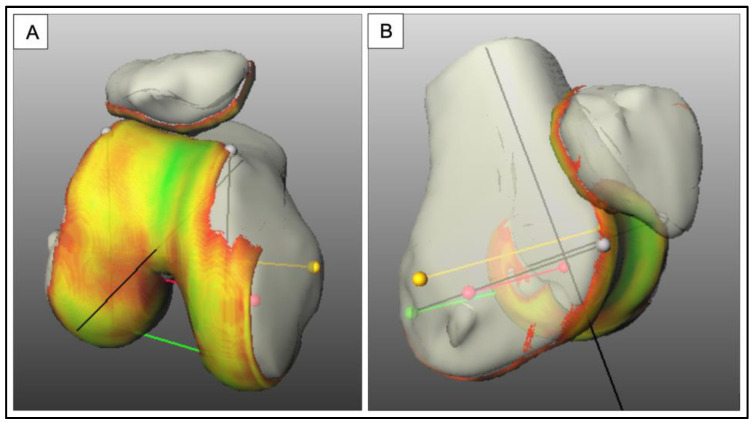
3D mesh of bone and cartilage structures to demonstrate the cartilage contact area (**A**,**B**), which serves as a basic matrix for the semi-automated determination of the kinematic parameters, yellow line: TEA (transepicondylar axis), red line: TCA (transcondylar axis), green line: PCA (posterocondylar axis). The color code shows the cartilage surface thickness. Green (5 mm layer thickness) over the gradient to yellow (2.5 mm layer thickness) and to red (1 mm layer thickness).

**Figure 4 jcm-12-01917-f004:**
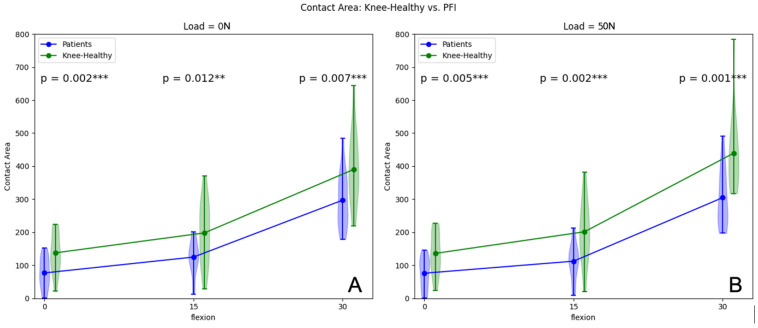
Patellofemoral CCA in unloaded (**A**) and loaded (50 N) situations (**B**) of patients and volunteers from 0° to 30° flexion. ** *p*-values < 0.05, *** *p*-values < 0.01.

**Figure 5 jcm-12-01917-f005:**
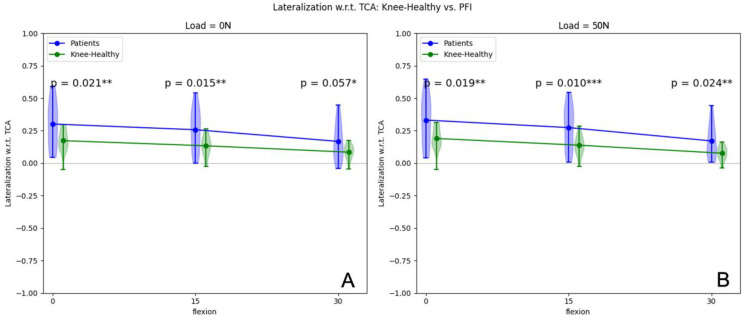
Patellar shift (mm) in unloaded (**A**) and loaded (50 N) situations (**B**) of patients and volunteers from 0° to 30° flexion. * *p*-values < 0.01, ** *p*-values < 0.05, *** *p*-values < 0.01.

**Figure 6 jcm-12-01917-f006:**
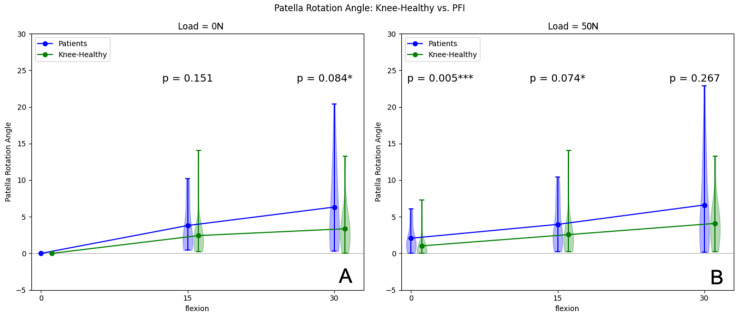
Patellar rotation (degree) in unloaded (**A**) and loaded (50 N) situations (**B**) of patients and volunteers from 0° to 30° flexion. * *p*-values < 0.01, *** *p*-values < 0.01.

**Table 1 jcm-12-01917-t001:** Overview of the patients’ and volunteers’ demographics.

	Patients	±SD	Volunteers	±SD	
Number of subjects	17		17		
Mean age (years)	26.47	±7.67	30.00	±5.81	
Mean height (cm)	174.59	±8.73	176.24	±9.50	
Mean weight (kg)	71.35	±10.88	75.47	±18.27	
Sex (female/male)	10/7	-	10/7	-	
Mean TEA distance (mm)	77.13	±6.05	79.63	±7.58	Matching reference (*p* = 0.245)

**Table 2 jcm-12-01917-t002:** Overview of measured parameters (mm^2^) of patients and volunteers in unloaded and loaded conditions.

			PFI Patients W/O Load Mean ± SD	Volunteers W/O Load Mean ± SD			PFI Patients 50 N Load Mean ± SD	Volunteers 50 N Load Mean ± SD	
**Flexion**		*n*=			***p*=**	*n*=			***p*=**
	CCA								
**0°**		17/17	76.65 ± 47.43	137.52 ± 59.58	**0.002**	17/17	75.91 ± 47.70	135.97 ± 63.09	**0.005**
**15°**		17/17	124.77 ± 52.71	197.97 ± 93.15	**0.012**	17/17	112.27 ± 55.07	201.31 ± 91.41	**0.002**
**30°**		17/17	297.11 ± 80.74	389.94 ± 108.05	**0.007**	17/17	304.73 ± 86.85	439.17 ± 115.29	**0.001**
	Shift								
**0°**		17/17	3.01 ± 1.81	1.73 ± 0.89	**0.021**	17/17	3.31 ± 1.93	1.90 ± 0.91	**0.019**
**15°**		17/17	2.57 ± 1.60	1.33 ± 0.75	**0.015**	17/17	2.73 ± 1.65	1.38 ± 0.76	**0.010**
**30°**		17/17	1.67 ± 1.41	0.84 ± 0.56	0.057	17/17	1.71 ± 1.30	0.76 ± 0.51	**0.024**
	Rotation								
**0°**		17/17	-	-	-	17/17	2.07 ± 1.89	1.02 ± 1.94	**0.005**
**15°**		17/17	3.80 ± 3.40	2.43 ± 3.14	0.151	17/17	3.96 ± 3.24	2.56 ± 3.19	0.074
**30°**		17/17	6.32 ± 5.88	3.36 ± 3.17	0.084	17/17	6.61 ± 6.55	4.09 ± 3.31	0.267

**Table 3 jcm-12-01917-t003:** Overview of parameters (mm^2^) without and with quadriceps activation (50 N).

			PFI Patients w/o Load Mean ± SD	PFI Patients 50 N Load Mean ± SD	Δ			Volunteers w/o Load Mean ± SD	Volunteers 50 n Load Mean ± SD	Δ	
**FLEX.**		*n*=				** *p* ** **=**	*n*=				** *p* ** **=**
	**CCA**										
**0°**		17	76.65 ± 47.43	75.91 ± 47.70	0.73 ± 0.27	0.979	17	137.52 ± 59.58	135.97 ± 63.09	−1.55 ± 3.51	0.758
**15°**		17	124.77 ± 52.71	112.27 ± 55.07	12.50± 2.36	0.309	17	197.97 ± 93.15	201.31 ± 91.41	3.35 ± −1.74	0.084
**30°**		17	297.11 ± 80.74	304.73 ± 86.85	7.62 ± 6.11	0.408	17	389.94 ± 108.05	439.17 ± 115.29	49.23 ± 7.24	**0.042**
	**Shift**										
**0°**		17	3.01 ± 1.81	3.31 ± 1.93	0.30 ± 0.01	**0.001**	17	1.73 ± 0.89	1.90 ± 0.91	0.17 ± 0.00	**0.030**
**15°**		17	2.57 ± 1.60	2.73 ± 1.65	0.16 ± 0.00	**0.010**	17	1.33 ± 0.75	1.38 ± 0.76	0.04 ± 0.00	0.156
**30°**		17	1.67 ± 1.41	1.71 ± 1.30	0.04 ±- 0.01	0.532	17	0.84 ± 0.56	0.76 ± 0.51	12 ± −0.01	0.352
	**Rot.**										
**0°**		17	-	2.07 ± 1.89	2.07 ± 1.89	-	17	-	1.02 ± 1.94	1.02 ± 1.94	-
**15°**		17	3.80 ± 3.40	3.96 ± 3.24	0.18 ± −0.18	0.379	17	2.43 ± 3.14	2.56 ± 3.19	1.16 ±2.04	0.246
**30°**		17	6.32 ± 5.88	6.61 ± 6.55	0.28 ± 0.67	0.717	17	3.36 ± 3.17	4.09 ± 3.31	2.07 ± 1.61	0.079

## Data Availability

All data are within the manuscript.

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
