# Peer review of "Change in Descriptive Kinematic Parameters of Patients with Patellofemoral Instability When Compared to Individuals with Healthy Knees—A 3D MRI In Vivo Analysis"

_jcm, 2023, doi:10.3390/jcm12051917_

Round 1
Reviewer 1 Report
The authors describe a nice way to examine patellofemoral kinematics in a group of patients with PFI compared to healthy. It is an interesting topic and of interest to clinicians and researchers. While the concepts of the paper are overall sound, I have a number of comments which I think need to be addressed.
line 57 – references are needed
Line 62 – I would suggest updating your references to inclide some of the more moderns studies. A quick literature search has found these two. Also, do that point, can you please clarify how your study will differ from what has been done already in-vivo?
Patellofemoral kinematics in patellofemoral pain syndrome: The influence of demographic factors - https://doi.org/10.1016/j.jbiomech.2021.110819
Dynamic Mediolateral Patellar Translation Is a Sex- and Size-Independent Parameter of Adult Proximal Patellar Tracking Using Dynamic 3 Tesla Magnetic Resonance Imaging - https://doi.org/10.1016/j.arthro.2021.10.014
Line 66 - why did you focus on the quads specifically where there are a number of other factors (which you have mentioned above) which might be important.
Methods
Was the patient recruitment from a consecutive series? As in, was every patient who met the inclusion criteria approached for the study?
Do you have any patient reported measures for these groups? Are the PFI group in more pain/have less function compared to the other group?
How did you settle on a sample size of 17?
Why did you focus on patients who were due for surgery? Do you think your results would have been different if you included patients who had instability but not on a surgical waitlist?
Line 103 – There seems to a difference between the groups in both age and weight? Do you think those differences influenced you results?
Did the controls have their MRI assessed for any pathology? While they were asymptomatic, they may still had some radiological features?
Line 129 – Why did you settle on 50N for quad loading? This is particularly important as the differences reports are quite small
What is the error/repeatability in your registration algorithm?
Statistics – Why did you chose the statistics test you used? It seems to me like a linear model may be more useful as you can control for a number of covariates. This may help to address some of the imbalance in the groups. You also need to take the measures over differences degrees into account. They are dependant on each other which the tests you ran do not account for
Results – while there are a number of significant differences reported, they are quite small. Do you think average differences 1-2 degrees are clinically meaningful
Can you comment on the variability seen in the patient group – there seems to be quite a large spread in this group
Discussion
You might need to temper your wording with being so definite on differences between groups. This is a small group with a number of covariates missing from the analyses.
Author Response
Point-by-point Answer Reviewer 1:
The authors describe a nice way to examine patellofemoral kinematics in a group of patients with PFI compared to healthy. It is an interesting topic and of interest to clinicians and researchers. While the concepts of the paper are overall sound, I have a number of comments which I think need to be addressed.
- I thank you very much for the positive review and the substantial input.
line 57 – references are needed
- We added the following references in line 57:
- Imhoff, F.B.; Funke, V.; Muench, L.N.; Sauter, A.; Englmaier, M.; Woertler, K.; Imhoff, A.B.; Feucht, M.J. The Complexity of Bony Malalignment in Patellofemoral Disorders: Femoral and Tibial Torsion, Trochlear Dysplasia, TT–TG Distance, and Frontal Mechanical Axis Correlate with Each Other. Knee Surg Sports Traumatol Arthrosc 2020, 28, 897–904, doi:10.1007/s00167-019-05542-y.
- Senavongse, W.; Amis, A.A. The Effects of Articular, Retinacular, or Muscular Deficiencies on Patellofemoral Joint Stability: A BIOMECHANICAL STUDY IN VITRO. The Journal of Bone and Joint Surgery. British volume 2005, 87-B, 577–582, doi:10.1302/0301-620X.87B4.14768.
Line 62 – I would suggest updating your references to inclide some of the more moderns studies. A quick literature search has found these two. Also, do that point, can you please clarify how your study will differ from what has been done already in-vivo?
Patellofemoral kinematics in patellofemoral pain syndrome: The influence of demographic factors - https://doi.org/10.1016/j.jbiomech.2021.110819
Dynamic Mediolateral Patellar Translation Is a Sex- and Size-Independent Parameter of Adult Proximal Patellar Tracking Using Dynamic 3 Tesla Magnetic Resonance Imaging - https://doi.org/10.1016/j.arthro.2021.10.014
- Thanks for the suggestions. Both studies are well known to me and are, in my opinion, appropriate and relevant for our present study. I have included both studies (references 12 and 13, see line 64).
Line 66 - why did you focus on the quads specifically where there are a number of other factors (which you have mentioned above) which might be important.
- Thank you for the question. To our knowledge, very little is known about the influence of quadriceps muscles in particular on patellofemoral contact in patients with PFI, which is why we focused of the influence of the Quadriceps, as a substantial part of this research. However, we agree with you that the influence of the mentioned and known factors is at least equally important and will be part of future projects. Though, we focused on the quadriceps in the present analysis due to the study design and the relatively small number of cases.
Methods
Was the patient recruitment from a consecutive series? As in, was every patient who met the inclusion criteria approached for the study?
- In fact, generally any patient in our outpatient clinic who met the inclusion criteria, for whom we had a suitable time to scheduled surgery, was approached regarding study participation. However, due to the complexity of the study design and the factor of time, the aforementioned 20 patients eventually emerged as potential subjects.
Do you have any patient reported measures for these groups? Are the PFI group in more pain/have less function compared to the other group?
- Thank you for the question. The patients with symptomatic PFI, as part of the inclusive criteria, have both more clinical complaints (pain and function) compared to the knee-healthy subjects. Unfortunately, the evaluation of these complaints is not part of the present study, but would be indeed very interesting and must be part of future projects.
How did you settle on a sample size of 17?
- The sample size of 17 was ultimately based on the complete records of the patellofemoral unstable patient cohort (after dropouts). A larger number for a statistically higher power would be desirable but is currently difficult to generate.
Why did you focus on patients who were due for surgery? Do you think your results would have been different if you included patients who had instability but not on a surgical waitlist?
- Thanks for the question. We think that patellofemoral instability is a very heterogeneous group of patients. In order to homogenize this group of patients, it seems to us appropriate to acquire the patients who are waiting for surgical stabilization.
We can imagine that patients who are suitable for a conservative therapy trial based on clinical experience also show an increased influence of quadriceps activity on patellofemoral kinematics. This "tool" for examining the influence of quadriceps muscles could be a prognostic factor for estimating the success of therapy in each case. However, this is completely hypothetical at this time.
Line 103 – There seems to a difference between the groups in both age and weight? Do you think those differences influenced you results?
- Thanks for the question. Comparing the age and weight of both cohorts (e.g. using students t.test), the present analysis shows no significant differences at n=17 (p=0.14; p=0.43). Nevertheless, weight in particular has recently been described as an influencing factor of patellofemoral kinematics. Therefore, in our opinion, there could conceivably be a difference between the two groups if the differences were larger.
Did the controls have their MRI assessed for any pathology? While they were asymptomatic, they may still had some radiological features?
- Thanks for the question. The preparation of the MRIs of the control subjects were made exclusively for the present study and did not show any "surprises" such as undiscovered arthrosis / major cartilage damage / ligament injuries.
Line 129 – Why did you settle on 50N for quad loading? This is particularly important as the differences reports are quite small
- Thanks for the question. We chose a load of 50N in order to activate the muscles on the one hand, but without generating enormous motion artifacts (since prospective motion correction also has a limit), in order to gain a robust and reproducible method. The standardized load also equalizes differences in the maximum strength of the quadriceps.
What is the error/repeatability in your registration algorithm?
- Thanks for your question, this is indeed an important point. One of our reasons for using automatic extraction of these parameters was to achieve robust results. To analyze the robustness of the entire setup, and not just the image registration, we did a robustness analysis in a previous study (https://onlinelibrary.wiley.com/doi/10.1002/jmri.26724 figure 5), where a subject was scanned three times for each loading/flexion situation. Then the algorithm was applied on these three sets of the scan and was compared with a manual segmentation and alignment. Although the subject was scanned in three different days, using our setup a good agreement was achieved among the scans.
Statistics – Why did you chose the statistics test you used? It seems to me like a linear model may be more useful as you can control for a number of covariates. This may help to address some of the imbalance in the groups. You also need to take the measures over differences degrees into account. They are dependant on each other which the tests you ran do not account for
- Thanks for your question. We chose Wilcoxon signed rank, because it is the suggested statistical test, when a limited number of samples is available. Another advantage of Wilcoxon signed rank is that it does not assume normality for the distribution of the data. We had intensively discussed the use of a linear model (e.g. regression model) with our statistical institutes and employees and we agree that this statistical evaluation of several covariates is certainly very suitable.
Results – while there are a number of significant differences reported, they are quite small. Do you think average differences 1-2 degrees are clinically meaningful
- Thank you for this question too. In our opinion, in order to be able to assess the clinical relevance of small differences, larger studies are required that examine kinematic differences using PROMs and other clinical outcome parameters (e.g. reluxation rate). I personally do not consider a difference of 1-2° to be groundbreaking, especially in the case of patella rotation.
Can you comment on the variability seen in the patient group – there seems to be quite a large spread in this group
- We consider the variability of the patient group most likely in the context of the heterogeneity of the patellofemoral instability. With the intention of homogenizing the patient group, only patients with low flexion PFI and planned surgical stabilization (using MPFL reconstruction) were included in the present analysis. Nevertheless, there still is the mentioned greater variability compared to the control group.
Discussion
You might need to temper your wording with being so definite on differences between groups. This is a small group with a number of covariates missing from the analyses.
- Thanks for this comment. We tried to temper our wording at several text passages, see Ll. 305, 311). It is also pointed out again that due to the small sample cohort no general conclusions can be drawn, since results for larger populations may vary.
Thank you very much for the valuable suggestions and the positive review.
Yours sincerely, Markus Siegel
Reviewer 2 Report
The reviewer applauds the authors for the conduction of the study. The authors investigated the change in descriptive kinematic parameters of patients with patellofemoral instability compared to individuals with healthy knees using 3D MRI in vivo analysis. The study has interesting viewpoint to answer the current clinical questions. There are some issues that should be addressed as listed below.
Title:
Title is well worded.
Introduction:
The introduction is comprehensive and succinct.
Materials and methods:
L92 and 95. What is the ‘technical issues’?
L121-123. It is unclear how was this device applied for the quadriceps muscle. It would be better to explain with illustration or photograph.
Figure 3. What the color indicate on the articular surface, red, yellow, and green? Please explain or add indicator for the color contour.
L175-176. ‘an anterior to posterior axis of the femur or patella’ What is the reference axis/line for this axis? Please explain or show in the figure.
Results:
The results are clearly written and explained.
Discussion:
Please briefly state the purpose of the CCA measurement.
Conclusion:
The conclusion is comprehensive.
Tables and figures are fine.
Reference list appears comprehensive.
Author Response
Point-by-point Answer
The reviewer applauds the authors for the conduction of the study. The authors investigated the change in descriptive kinematic parameters of patients with patellofemoral instability compared to individuals with healthy knees using 3D MRI in vivo analysis. The study has interesting viewpoint to answer the current clinical questions. There are some issues that should be addressed as listed below.
- Thank you very much for your positive review.
Title:
Title is well worded.
- Thank you.
Introduction:
The introduction is comprehensive and succinct.
- Thank you.
Materials and methods:
L92 and 95. What is the ‘technical issues’?
- Thanks for your question. The "technical issues" amounted to a defective MRI coil of the 3 Tesla MRI and a failure of the pneumatic pump of the loading device, both of which could be repaired promptly.
L121-123. It is unclear how was this device applied for the quadriceps muscle. It would be better to explain with illustration or photograph.
- Thank you for the note. In lines 128-131 the attachment of the loading device, which leads to quadriceps activation, is described. Figure 2 shows the experimental setup including the loading apparatus. I hope this explanation will suffice.
Figure 3. What the color indicate on the articular surface, red, yellow, and green? Please explain or add indicator for the color contour.
- The color code shows the cartilage surface thickness. Green (5mm layer thickness) over the gradient to yellow (2.5mm layer thickness) and to red (1mm layer thickness) (see lines 158-159). If you want me, I can provide an indicator bar.
L175-176. ‘an anterior to posterior axis of the femur or patella’ What is the reference axis/line for this axis? Please explain or show in the figure.
- Thanks for the question. We defined the Patellar rotation as the angle between the medial-lateral axis of the patella and the femoral transepicondylar axis (TEA), projected onto the coronal plane of the femur. We added this information to the lines 178-180.
Results:
The results are clearly written and explained.
- Thank you.
Discussion:
Please briefly state the purpose of the CCA measurement.
Conclusion:
The conclusion is comprehensive.
- Thank you.
Tables and figures are fine.
Reference list appears comprehensive.
Thank you very much for the valuable comments and the positive review.
Best regards,
Markus Siegel
Round 2
Reviewer 1 Report
I'd like to thank the authors for considering my comments and providing thoughtful responses. However, most of these responses were not encorportated into the text, which is what I would expect. You have added in the additional references, but alot of the clarification you provided in your response to my questions is missing. I think it would help the readibility/scientific rigour of this paper if, where possible, you could integrate these responses.
Author Response
VERSION 2:
Point-by-point Answer Reviewer 1:
The authors describe a nice way to examine patellofemoral kinematics in a group of patients with PFI compared to healthy. It is an interesting topic and of interest to clinicians and researchers. While the concepts of the paper are overall sound, I have a number of comments which I think need to be addressed.
- I thank you very much for the positive review and the substantial input.
line 57 – references are needed
- We added the following references in line 57:
- Imhoff, F.B.; Funke, V.; Muench, L.N.; Sauter, A.; Englmaier, M.; Woertler, K.; Imhoff, A.B.; Feucht, M.J. The Complexity of Bony Malalignment in Patellofemoral Disorders: Femoral and Tibial Torsion, Trochlear Dysplasia, TT–TG Distance, and Frontal Mechanical Axis Correlate with Each Other. Knee Surg Sports Traumatol Arthrosc 2020, 28, 897–904, doi:10.1007/s00167-019-05542-y.
- Senavongse, W.; Amis, A.A. The Effects of Articular, Retinacular, or Muscular Deficiencies on Patellofemoral Joint Stability: A BIOMECHANICAL STUDY IN VITRO. The Journal of Bone and Joint Surgery. British volume 2005, 87-B, 577–582, doi:10.1302/0301-620X.87B4.14768.
Line 62 – I would suggest updating your references to inclide some of the more moderns studies. A quick literature search has found these two. Also, do that point, can you please clarify how your study will differ from what has been done already in-vivo?
Patellofemoral kinematics in patellofemoral pain syndrome: The influence of demographic factors - https://doi.org/10.1016/j.jbiomech.2021.110819
Dynamic Mediolateral Patellar Translation Is a Sex- and Size-Independent Parameter of Adult Proximal Patellar Tracking Using Dynamic 3 Tesla Magnetic Resonance Imaging - https://doi.org/10.1016/j.arthro.2021.10.014
- Thanks for the suggestions. Both studies are well known to me and are, in my opinion, appropriate and relevant for our present study. I have included both studies (references 12 and 13, see line 64).
Line 66 - why did you focus on the quads specifically where there are a number of other factors (which you have mentioned above) which might be important.
- Thank you for the question. To our knowledge, very little is known about the influence of quadriceps muscles in particular on patellofemoral contact in patients with PFI, which is why we focused of the influence of the Quadriceps, as a substantial part of this research. However, we agree with you that the influence of the mentioned and known factors is at least equally important and will be part of future projects. Though, we focused on the quadriceps in the present analysis due to the study design and the relatively small number of cases.
- We added „In particular, little is known about the influence of the quadriceps on patellofemoral kinematics and the patellofemoral contact.“ In Ll 63 and 64
Methods
Was the patient recruitment from a consecutive series? As in, was every patient who met the inclusion criteria approached for the study?
- In fact, generally any patient in our outpatient clinic who met the inclusion criteria, for whom we had a suitable time to scheduled surgery, was approached regarding study participation. However, due to the complexity of the study design and the factor of time, the aforementioned 20 patients eventually emerged as potential subjects.
- In LL 93-94 we can find the description of the 20 suitable candidates for he study.
Do you have any patient reported measures for these groups? Are the PFI group in more pain/have less function compared to the other group?
- Thank you for the question. The patients with symptomatic PFI, as part of the inclusive criteria, have both more clinical complaints (pain and function) compared to the knee-healthy subjects. Unfortunately, the evaluation of these complaints is not part of the present study, but would be indeed very interesting and must be part of future projects.
- To accentuate the clinical difference between the groups, in Ll 80 "clinical apparent" and in Ll 84-85 "Only volunteers without a history of knee pain were included."
How did you settle on a sample size of 17?
- The sample size of 17 was ultimately based on the complete records of the patellofemoral unstable patient cohort (after dropouts). A larger number for a statistically higher power would be desirable but is currently difficult to generate.
- We added: „The sample size of 17 was ultimately based on the complete records of the patellofemoral unstable patient cohort (after dropouts).“ In LL 99-101.
Why did you focus on patients who were due for surgery? Do you think your results would have been different if you included patients who had instability but not on a surgical waitlist?
- Thanks for the question. We think that patellofemoral instability is a very heterogeneous group of patients. In order to homogenize this group of patients, it seems to us appropriate to acquire the patients who are waiting for surgical stabilization.
- We can imagine that patients who are suitable for a conservative therapy trial based on clinical experience also show an increased influence of quadriceps activity on patellofemoral kinematics. This "tool" for examining the influence of quadriceps muscles could be a prognostic factor for estimating the success of therapy in each case. However, this is completely hypothetical at this time.
- We added „To ensure relative homogenization of the trial cohort in the heterogeneous condition of patellofemoral instability, only patients with low flexion PFI were included and all patients were recruited from the waiting list for patellofemoral stabilizing surgery. It is possible that results may vary in patients who were not preselected in this regard.“ In the limitation section LL 377-381
Line 103 – There seems to a difference between the groups in both age and weight? Do you think those differences influenced you results?
- Thanks for the question. Comparing the age and weight of both cohorts (e.g. using students t.test), the present analysis shows no significant differences at n=17 (p=0.14; p=0.43). Nevertheless, weight in particular has recently been described as an influencing factor of patellofemoral kinematics. Therefore, in our opinion, there could conceivably be a difference between the two groups if the differences were larger.
- Since there is no statistical relevance here, we have decided not to create an extra note.
Did the controls have their MRI assessed for any pathology? While they were asymptomatic, they may still had some radiological features?
- Thanks for the question. The preparation of the MRIs of the control subjects were made exclusively for the present study and did not show any "surprises" such as undiscovered arthrosis / major cartilage damage / ligament injuries.
- „(…)and were also examined for study purposes according to the MRI protocol.“ was added in L98.
Line 129 – Why did you settle on 50N for quad loading? This is particularly important as the differences reports are quite small
- Thanks for the question. We chose a load of 50N in order to activate the muscles on the one hand, but without generating enormous motion artifacts (since prospective motion correction also has a limit), in order to gain a robust and reproducible method. The standardized load also equalizes differences in the maximum strength of the quadriceps.
- We added: “We chose a load of 50N in order to activate the muscles without generating enormous motion artifacts, in order to gain a robust and reproducible method” In LL 135-137
What is the error/repeatability in your registration algorithm?
- Thanks for your question, this is indeed an important point. One of our reasons for using automatic extraction of these parameters was to achieve robust results. To analyze the robustness of the entire setup, and not just the image registration, we did a robustness analysis in a previous study (https://onlinelibrary.wiley.com/doi/10.1002/jmri.26724 figure 5), where a subject was scanned three times for each loading/flexion situation. Then the algorithm was applied on these three sets of the scan and was compared with a manual segmentation and alignment. Although the subject was scanned in three different days, using our setup a good agreement was achieved among the scans.
- The study mentioned is cited several times throughout the paper, but if you wish we can add a section and explicitly refer to it again.
Statistics – Why did you chose the statistics test you used? It seems to me like a linear model may be more useful as you can control for a number of covariates. This may help to address some of the imbalance in the groups. You also need to take the measures over differences degrees into account. They are dependant on each other which the tests you ran do not account for
- Thanks for your question. We chose Wilcoxon signed rank, because it is the suggested statistical test, when a limited number of samples is available. Another advantage of Wilcoxon signed rank is that it does not assume normality for the distribution of the data. We had intensively discussed the use of a linear model (e.g. regression model) with our statistical institutes and employees and we agree that this statistical evaluation of several covariates is certainly very suitable.
- We think in this regard no adjustment in the text is necessary
Can you comment on the variability seen in the patient group – there seems to be quite a large spread in this group
- We consider the variability of the patient group most likely in the context of the heterogeneity of the patellofemoral instability. With the intention of homogenizing the patient group, only patients with low flexion PFI and planned surgical stabilization (using MPFL reconstruction) were included in the present analysis. Nevertheless, there still is the mentioned greater variability compared to the control group.
- We have now commented on this in the Limitations (Ll 377 - 381)
Discussion
You might need to temper your wording with being so definite on differences between groups. This is a small group with a number of covariates missing from the analyses.
- Thanks for this comment. We tried to temper our wording at several text passages, see Ll. 305, 311). It is also pointed out again that due to the small sample cohort no general conclusions can be drawn, since results for larger populations may vary.
Thank you very much for the valuable suggestions and the positive review.